# Inferring transportation mode from smartphone sensors: Evaluating the potential of Wi-Fi and Bluetooth

**Andreas Bjerre-Nielsen**[1,2]*, **Kelton Minor**[1,3], **Piotr Sapieżyński**[4], **Sune Lehmann**[1,5], **David Dreyer Lassen**[1,2]

**1** Copenhagen Center for Social Data Science, University of Copenhagen, Copenhagen, Denmark, **2** Department of Economics, University of Copenhagen, Copenhagen, Denmark, **3** Department of the Built Environment, Aalborg University, Copenhagen, Denmark, **4** Khoury College of Computer Sciences, Northeastern University, Boston, Massachusetts, United States of America, **5** DTU Compute, Technical University of Denmark, Lyngby, Denmark

* andreas.bjerre-nielsen@econ.ku.dk

**Data Availability Statement:** The data for estimating and evaluating the model is available along with the manuscript in an anonymized format in its Supporting Information files. The additional

## Abstract

Understanding which transportation modes people use is critical for smart cities and planners to better serve their citizens. We show that using information from pervasive Wi-Fi access points and Bluetooth devices can enhance GPS and geographic information to improve transportation detection on smartphones. Wi-Fi information also improves the identification of transportation mode and helps conserve battery since it is already collected by most mobile phones. Our approach uses a machine learning approach to determine the mode from pre-prepocessed data. This approach yields an overall accuracy of 89% and average $F_1$ score of 83% for inferring the three grouped modes of self-powered, car-based, and public transportation. When broken out by individual modes, Wi-Fi features improve detection accuracy of bus trips, train travel, and driving compared to GPS features alone and can substitute for GIS features without decreasing performance. Our results suggest that Wi-Fi and Bluetooth can be useful in urban transportation research, for example by improving mobile travel surveys and urban sensing applications.

## 1 Introduction

Transportation departments and urban researchers have long aimed to model transportation behavior, measure how people traverse time and space, and understand the factors that predict travel-related decisions [1]. World-wide threats such as global warming, planning level problems such as sprawl, transportation system issues such as congestion, and environmental health concerns such as pollution are all connected to peoples' mode choices, especially the use of automobiles [2–5]. The link between the ways we get around and the costs these actions create has recently compelled cities to promote self-powered mobility options such as walking and biking [6] and develop more sustainable forms of transportation including public transit, such as bus and rail [7]. However, cities frequently lack the tools to evaluate these initiatives

data on students cannot be shared publicly because of privacy concerns. Data are available from the Steering Committee of the Copenhagen Networks Study (contact via ddl@econ.ku.dk) for researchers who meet the criteria for access to confidential data.

**Funding:** ABN, DDL, KRM, SL acknowledge funding from Kraks Fond - Institute for Urban Economic Research (URL: https://kraksfondbyforskning.dk/en/). The funders had no role in study design, data collection and analysis, decision to publish, or preparation of the manuscript.

**Competing interests:** The authors have declared that no competing interests exist.

and often resort to simple surveys and manual traffic counts which are resource intensive and only capture momentary behavior. At the same time, technological advances have furnished citizens, researchers and infrastructures with new mobile instruments, such as smartphones and activity trackers, to monitor and respond to human behavior [8–10]. These applications include the use of geospatial data to help people navigate through the environment and decide which modes of transit to take [11], track human health and behavior such as personal fitness [12–14], record personal exposures to hazards and environmental toxins [15–18], track individual impacts on the environment [19], and use location based services [20–22]. Inaccuracies in such predictions can cause people to lose their way, introduce noise into exposure estimates, and compromise the promise of 'smart' infrastructure [23].

In previous research, a variety of built-in smartphone sensors have been used to estimate the transportation mode of mobile individuals, with studies most commonly relying on GPS and accelerometer data [24]. Other studies have leveraged the physical environment around users by using contextual clues from geographic information systems (GIS), e.g. railroad network information, to help distinguish between modes with similar velocities and accelerations [25]. Other research has emphasized the potential utility of magnetometer, barometer and other remote sensor technologies. [26]. Recent transportation inference approaches, however, do not widely use a now ubiquitous layer of context: Wi-Fi and Bluetooth traces.

In this paper, we describe a classification system based on Wi-Fi and Bluetooth signals, in conjunction with individual location data and geospatial context information, to improve inferences of transportation behavior. We distinguish among self-powered transportation vs. use of car vs. public transport usage as transportation modes. Our main contribution is to extend the previous work [19] by using the presence of network names (i.e. service set identifiers, abbreviated SSIDs) broadcasted exclusively by Wi-Fi routers placed on public transportation vehicles. We make two additional contributions, (i) inferring location through presence of detected Wi-Fi routers, and (ii) deriving additional features to help with identification of transportation mode from the presence and churn of nearby Wi-Fi and Bluetooth devices. We show that combining our novel features with user location data in a machine learning model improves upon methods based on location data alone. To inspect how our model scales to a larger population, we apply it to a mobility dataset of trips from over 800 individuals collected over two years and evaluate descriptive measures. Based on our results, we discuss advantages of using Wi-Fi scans for transportation inference, including low power consumption and accessible semantic information about a user's exposure to public transportation modes. Likewise, we discuss the empirical limitations of our research and recommend that future studies investigate the efficacy of Wi-Fi features across other geographic contexts and alongside other sensor information such as accelerometry.

The paper proceeds as follows. In Section 2 we review existing research on transportation mode detection. Section 3 contains an overview of our approach for inferring transportation method from pre-processed data. Section 4 describes the procedure for data collection, the underlying hardware and software used, as well as the pre-processing of raw data from Wi-Fi, Bluetooth and location measures. In Section 5 we train, test, and evaluate a random forest classifier using real life labelled data. Finally, we conclude the paper in Section 6.

## 2 Related work

Past research on travel patterns has investigated various recording technologies and classification methods to infer people's mobility behaviors and transportation modes. In this first section we provide a brief overview of existing methods to identify individual transportation behaviors and areas where location data as well as Wi-Fi and Bluetooth traces, along with

contextual information about the transportation network and urban structure, can contribute to trip classification.

Early transportation research was often conducted by national agencies and used phone-based interviews, mail-in surveys, or travel diaries to estimate transportation patterns. These self-report methods are subject to respondent fatigue, incompleteness, inaccuracies, recency effects, and under-reporting of brief or compound trips that are less easily remembered [27–29]. Adding to concerns about the viability of travel surveys, declining response rates have long been observed [30, 31].

Limitations of travel surveys and the advent of smartphones have prompted researchers to utilize mobile phones in transportation research. By extracting information from mobile phones researchers can improve the ecological validity of transportation models [32]. Compared to other methods of measuring human mobility that capture information about a single mode of transportation, continuously collected mobile phone data can record traces of an individual's travel trajectories over multiple days and across multiple modes of transportation [33]; see recent reviews in refs. [34, 35]. Some researchers have used cell tower networks and end point localization over time to infer transportation mode. However, access to cell tower information is limited by many APIs which only report the currently connected tower or often none at all [36]. In Refs. [37, 38], signals from cell tower trilateration were used to estimate device positions and identify the home location of mobile phone users. These location estimates, however, lacked spatial precision with an uncertainty of around 320 m—inadequate for detecting, for example, short trips within a neighborhood.

Previous research on mobility inference has employed GPS signals to extract features such as average and maximal speed during transportation in order to predict both stop locations and transportation modes [39–41]. Unfortunately, the accuracy of GPS diminishes substantially in an indoor setting [42]. In Ref. [43] it was found that GPS alone was inadequate for registering most train and tram trips in urban environments in the Netherlands. Researchers exclusively relying on GPS to infer urban travel behavior have also had to discard large portions of the data to contend with poor GPS reception due to signal deflection off of tall buildings and imprecise pedestrian localization due to slow movements obscured by these deflections [44]. Given that a growing majority of the human population lives in urban areas [45] and spend a majority of their time indoors, GPS alone provides insufficient information to infer localized transportation and dwelling behavior without additional processing of routes or complementary sensors [46].

Several studies have augmented GPS traces with contextual GIS information about relevant adjacencies, including proximity to public transportation stops and networks to help classify mobility data with mode of transportation [47, 48]. Ref. [25] found that including contextual transit network features such as the distance from a user's measured location to rail lines, bus stops and real-time bus locations markedly improves the precision of detecting motorized car and bus modes as well as biking. Without such contextual information, these modes are commonly confused since speeds can be similar regardless of whether someone is biking, driving, or taking a bus, especially in cities during periods of peak congestion. These combined location data-contextual information approaches rely on high frequency GPS sampling and a constant stream of external GIS information processed on a central server which can be both computationally and battery intensive for everyday mobile sensing applications and location based services [49]. Other research has established the high quality of infrastructure data available from the Google Maps API [50]. The API has been leveraged as a data source for road networks [25] and recently to distinguish biking paths from roads for driving [51].

Here we show that the accuracy of transportation mode prediction can be further increased by complementing other mobile sensing methods with crowdsourced Wi-Fi scan

information about a smartphone's surrounding access points (APs). Utilizing Wi-Fi provides access to several new features such as the longest visible router, change in the number of visible routers, and variation in signal strength of routers [19, 36]. Ref. [42] used the received signal strength (RSSI) of Wi-Fi AP scans and a geolocation service based on these scans to detect dwelling behaviors more accurately than with GPS alone, but did not classify mobile transportation states. Conversely, [52] used the variance in Wi-Fi RSSI measurements to infer whether people were moving but did not attempt to classify specific modes of urban transport between buildings. Ref. [53] combined cell tower information and Wi-Fi to distinguish between stationary, walking, and driving modes with 88% accuracy but excluded the often confused modes of bus, bike, and other forms of public transportation that frequently exhibit similar travel speeds as driving [54]. Important earlier work in this context concludes that Wi-Fi based RSSI features alone work well for coarse grained transportation classification between still and mobile modes, but struggle to distinguish fine grained differences between modes with similar speeds [24]. See Table 1 for an overview of model accuracy of existing work using Wi-Fi for mode classification. Furthermore, others point out that Wi-Fi RSSI approaches have only been demonstrated with high scanning frequencies of at least once every three seconds [55]. It is therefore unclear whether these results extend to the low scanning frequencies adopted by most mobile phones. Finally, while the previous Wi-Fi based studies collected data through active solicitation of consenting mobile phone users, recent research has explored whether population-level dwelling and coarse mobility activity can be inferred passively using Wi-Fi probe or smart card data [56–60]. However, most mobile phones now randomize their MAC addresses to protect privacy, limiting the viability of Wi-Fi probe solutions to reliably track individual-level transportation decisions over time [61]. Similarly, although smart cards and other big data sources like taxi use [62, 63] provide macro-level information about specific transit types, they do not account for individual-level behaviors beyond the participating modes [64, 65].

Here we argue, however, that previous mobility studies have yet to fully utilize the information contained in the SSIDs of Wi-Fi access points scanned from mobile devices. For example SSIDs can directly reveal the mode of transportation, e.g. buses in the Greater Copenhagen area have free Wi-Fi access points on board named "Bedrebustur" which translates to "A better bus trip" (see Fig 1). Similarly, all local trains in Copenhagen contain Wi-Fi access points with the SSID "Stog Wifi". A similar naming convention is used in inter-city trains running between regions. Additionally, researchers outside of the transportation community have shown that mobile Wi-Fi traces can be used in conjunction with concurrent mobile GPS signals to create a crowdsourced map of Wi-Fi router locations [66]. Since the Android mobile operating system already schedules regular Wi-Fi scans, this additional information can be paired with mobile GPS data to increase the number of location samples without imparting additional cost to the battery [67].

Similar to Wi-Fi sensors, accelerometers are also ubiquitous in most smart phones and require less energy to operate than GPS [68]. Although accelerometer-based classification has become a fixture in major phone operating systems (see for instance the Android OS

**Table 1. An overview of related research on Wi-Fi based transportation detection.**

| Reference | Sensors | Modes | Users | Time | Accuracy |
|---|---|---|---|---|---|
| [52] | Wi-Fi | Moving, Still | 2 | 12 hrs | 92pct |
| [19] | Wi-Fi, GSM | Walk, Motor, Still | 2 | 13 hrs | 83pct |
| [24] | Wi-Fi, GPS | Walk, Run, Bike, Motor, Still | 16 | 120 hrs | 79pct |

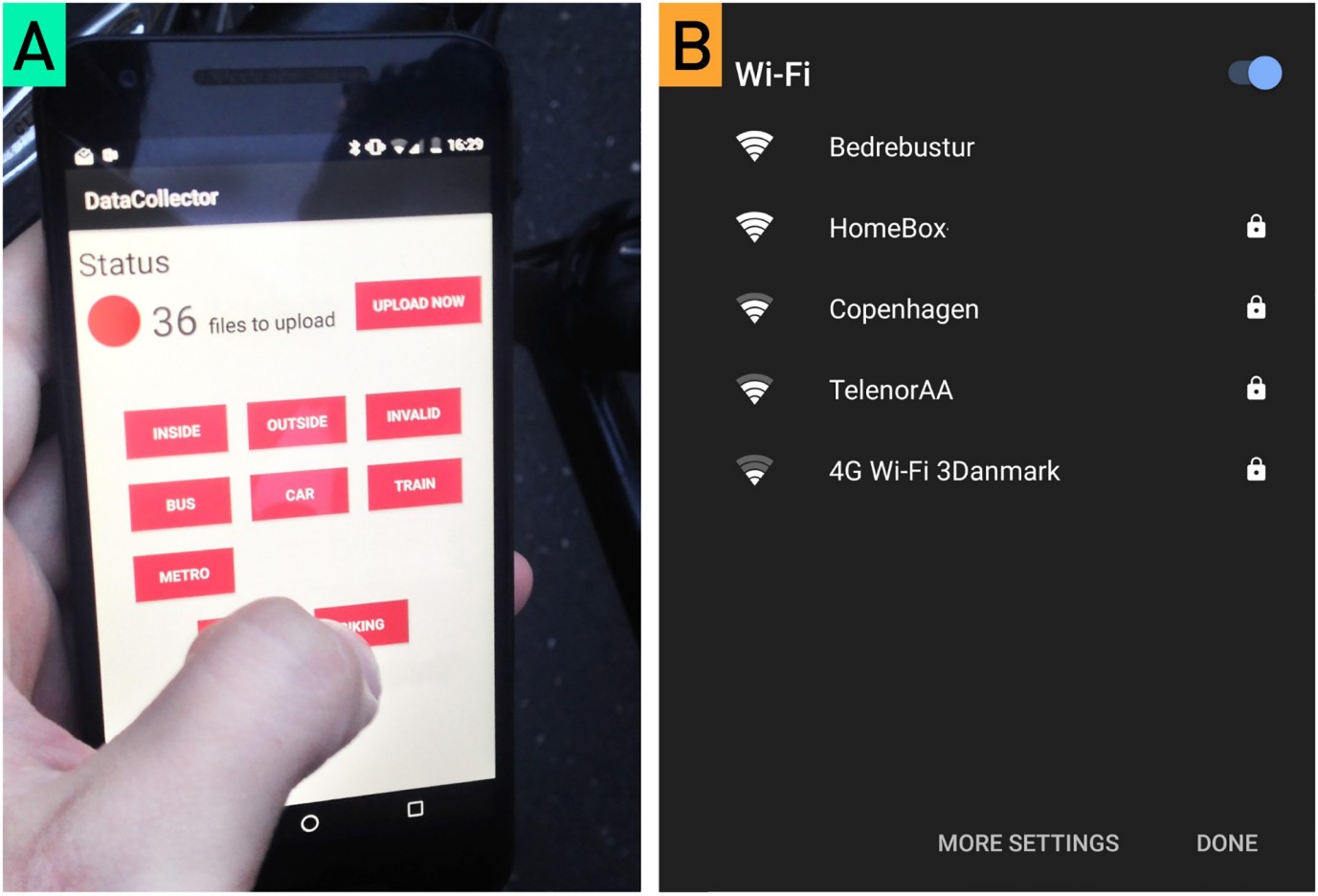

**Fig 1. Mobile travel diary app and results of a Wi-Fi scan.** (A) The in-situ travel diary used by participants to record their transportation mode decisions while they were on the move. (B) An example of how the Wi-Fi SSIDs associated with certain modes of public transportation can be readily collected by crowdsourced Wi-Fi scans from mobile phones.

implementation [69]), and has advanced the state-of-the-art of standalone location based services and transportation classification methods [68, 70], it neither provides semantic clues about public transportation modes nor spatial information without the concurrent use of GPS. Hence, the Wi-Fi based approach taken in this paper is tailored towards transportation science applications where either further discrimination between public and other modes is desired or accelerometer data is unavailable. A portion of the data in this paper (see Results section 5.2) originates from a large-scale mobile phone-based data collection undertaken as a part of the Copenhagen Networks Project [71, 72]. The data collector app employed in this project recorded sparse mobile sensor data from GPS and existing Wi-Fi scans but did not collect high frequency accelerometer data to conserve battery life.

## 3 Methods

The collected data has been authorized by Danish Data Protection Agency, with #2012-41-0664. In terms of privacy all participants have provided full consent for use of their data for research purposes. For the first dataset, used for training the model, oral consent was provided

to also share the data. We obtained written consent from participants in the Copenhagen Networks Study. At any time a student could exit the study and request to have their data deleted.

To infer the mode of transportation we constructed models to classify users' transportation mode. The models were estimated and evaluated on annotated travel journey data using preprocessed data from smartphone sensors (see the next section for a description of the data). The models were estimated jointly after data collection on a computer.

We built the models for classification using a supervised machine learning approach. The models we explore here were selected among common supervised machine learning models for classification problems. We note that other more powerful methods, e.g. variations of artificial neural networks, might yield improved performance but also require longer time for estimation [73] and do not permit us to interpret features.

Our modelling approach consists of two overall steps: i) model building and ii) measuring model performance. The model building phase included both selecting the optimal hyperparameter(s) and estimating the classifier (i.e. classification model). The pipeline for model building consisted of five steps outlined below. Each of these steps was estimated on the dataset for training the model and never in the data for validating the model.

1. Load pre-processed data

2. Impute missing values using the mean of non-missing values.

3. Standardize the data (zero mean, unit standard deviation).

4. Perform Recursive Elimination of Features (RFE) using the classification model. This process consists of removing the least relevant feature one at a time until the desired number of features was reached (i.e. as specified by the hyperparameters). Thus, the hyperparameters in this step are: number of features to keep during RFE and classification model's hyperparameters used during RFE.

5. Re-estimate the classifier after RFE. The hyperparameters in this step are the classification model's hyperparameters that remain after RFE. See S1 Appendix for an overview of all hyperparameters.

We limited the number of classification models to three: i) random forest (*RF*) [74]; ii) multinomial logistic regression (*LR*) as pioneered by McFadden [1]; and iii) support vector machine (*SVM*) for classification with a linear kernel [75]. This choice of widely-adopted models is consistent with the prior literature [25, 76]. We used versions of these models implemented in the scikit-learn package (version 0.20) for Python [77, 78].

We specify RF as the primary model in our analysis. The implementation of RF is based on Classification and Regression Trees (CART) [79]. RF relies on estimating multiple decision trees on different, random bootstrap samples of the data. Each decision tree attempts to split the data using features into subsets that contain a high share of a certain transportation mode [79]. The procedure of combining the trees is known as bagging and often raises the performance by relying on multiple models, rather than a single [79]. Further description of our models along with their hyperparameters can be found in the S1 Appendix.

The supervised machine learning models were estimated on subsets of the data ("training data"). After each model estimation was completed, we applied the model to a separate subsample ("test data"), to measure how the model generalizes to new data. Estimation of the models was performed at the minute level. When evaluating the models we only report performance at the segment level. Note that S1 Table contains a comprehensive list of model performance metrics at both the minute and segment level.

For model evaluation and choosing hyperparameters we use the model accuracy. We calculate overall model accuracy as the share of true labels assigned. For each specific transportation mode we also measure accuracy with each of the following: recall (the share of true positive labels assigned relative to the actual count of observations for the mode) and precision (the share of true labels assigned relative to the predicted count of observations for the mode [80]). We also use the $F_1$ score which is defined as $F_1 = \frac{2pr}{p+r}$ where $p$ is precision and $r$ is recall.

Since our dataset is limited in size, we use cross validation (CV) to estimate the performance of our machine learning model on unseen data. In order to measure the uncertainty of the model performance, we therefore apply "nested resampling" [81] which is a modification of standard CV. In standard CV the data is split at random into a number of training and test sets (folds). In nested resampling, the training data is further split into parts for training and validation, which is called the inner CV step. This inner loop is used to select the optimal hyperparameters based on out-of-sample model performance. The use of nested resampling—where we draw the training data as a subsample of the original training fold of the data—provides a better estimate of generalization error than only performing the split once [81]. Compared to drawing samples from the data for training the model using the bootstrap method, subsampling, i.e. sampling without replacement, is less prone to model over-fitting since there are no repeated observations [82].

An overview of the model building approach is found below. For each combination of classification model, feature set and target variable, the model building and validation consisted of the following steps:

- *Outer loop: subsampling*: Split data randomly into training and test data sets of respectively 75% and 25%. We split the data 1000 times with each sample drawn independently. The split into training and test data was performed at the segment level to ensure that no parts of a segment were used in both training and test data. Our splits were balanced such that test and training had the same proportion of each mode. Note that the split was performed for each set of the set of transportation features we modelled, i.e. for 2, 3, and 5 modes since these shared the same model target, i.e. predicted outcome.

  - *Inner loop: k-fold cross-validation*: The training data was then (once more) split into four (non-overlapping) folds of validation data, each with 25% of the data. This implies that unlike the outer CV the inner CV is not drawn by resampling independently. Instead the inner CV splits the data evenly into the four bins as in standard CV. For each fold the remaining 75% was used for training data. Like the outer loop, the splitting of data was balanced and split at the segment level. The model was estimated once for each combination of training fold in the inner CV and each hyperparameter.

- *Optimal model*: We selected the combination of hyperparameters that maximized the mean out-of-sample overall recall on the inner CV validation data sets. Using the entire training set from the outer loop the model was then re-estimated using the optimal hyperparameters.

- *Performance*: We evaluated the optimal model by computing the $F_1$ score, overall accuracy as well as precision and recall out-of-sample on the test dataset (of the outer-loop).

We use the resampled model performances to compute tests of whether two models have the same level of performance. Taking one of the two models as the null, we compute the $p$-value using the corrected resampled $t$-test [83]. This allows the standard $t$-test to be computed using a correction on the estimated variance.

## 4 Data

### 4.1 Collection of data

Similar to other relevant and recent work, the ground truth data was collected by a select number of users, across a variety of modes, over an extended period of time [19, 68]. In our study, four participants recorded their daily transportation behaviors for a combined 119 hours of transportation and 527 trip segments spanning over 158 days. The data we analyze below has been categorized into the following modes by the participants: self-powered (i.e. walk or bike), bus, drive, train, and metro. An overview of the total data used by mode of transportation can be found in Table 2. The study area was limited to Zealand, Denmark; see a map in Fig 2. Full consent were provided by all participants for research purposes. The consent was provided both for participants (oral) and students participating in Sensible DTU [71] (written).

**Software.** In order to acquire high quality ground truth data for fine-grained modes of transportation (metro, train, bus, car, etc.), we used in-situ travel diaries. To improve upon the accuracy of previous ground truth methods that relied on post-hoc self-report data subject to recall bias, imprecision, and omission, we devised a mobile 'transportation journal' app that permitted the participants to log their detailed transportation behaviors in real time on their cell phone (see Fig 1A). To increase ease of use, we only required participants to log the transition points between modes (i.e. boarding a bus, parking a bike, exiting a train). Initial pilot tests showed that the participants occasionally forgot to label all of their trips. We addressed this issue by displaying the time when the latest entry was created. Furthermore, we added an "invalid" button for the participants to label data for exclusion if they had forgotten to label one or more previous mode changes.

**Hardware and sensors.** We employed Android phones using a custom app for data collection. The app collected location, Bluetooth, and Wi-Fi as detailed in [71]. The location data consisted of GPS coordinates as well as user and time information. The location data was retrieved through the Google Location API which uses a mix of GPS, Wi-Fi, and nearby cell phone towers [71]. The Wi-Fi and Bluetooth data consisted of the available names for scanned routers or devices (SSID) as well as their unique identifiers known as MAC-addresses. Furthermore, these data contained information about the user ID as well as the timestamp and signal strength of the scan. The relevant phone sensors, i.e. Bluetooth, location, and Wi-Fi, were set to register a measurement at least every five minutes. Additionally, the app collected the results of scans requested by all other 3rd party and system applications; this resulted in a median scan interval of 15 seconds for Wi-Fi. Data from the phones' accelerometers were not included to minimize consumption of battery.

### 4.2 Pre-processing of data

The pre-processing of the data consisted of three steps. First, using raw data from Wi-Fi and location traces we computed a measure of combined location data at 1 min. resolution.

**Table 2. Data sample overview organized by transportation mode.** "Self-powered" refers to walking, running, and biking.

|  | Count of segments | Total data hours |
|---|---|---|
| Self-powered | 346 | 65.4 |
| Car | 67 | 25.0 |
| Bus | 67 | 16.6 |
| Metro | 24 | 4.2 |
| Train | 23 | 7.4 |
| Total | 527 | 118.6 |

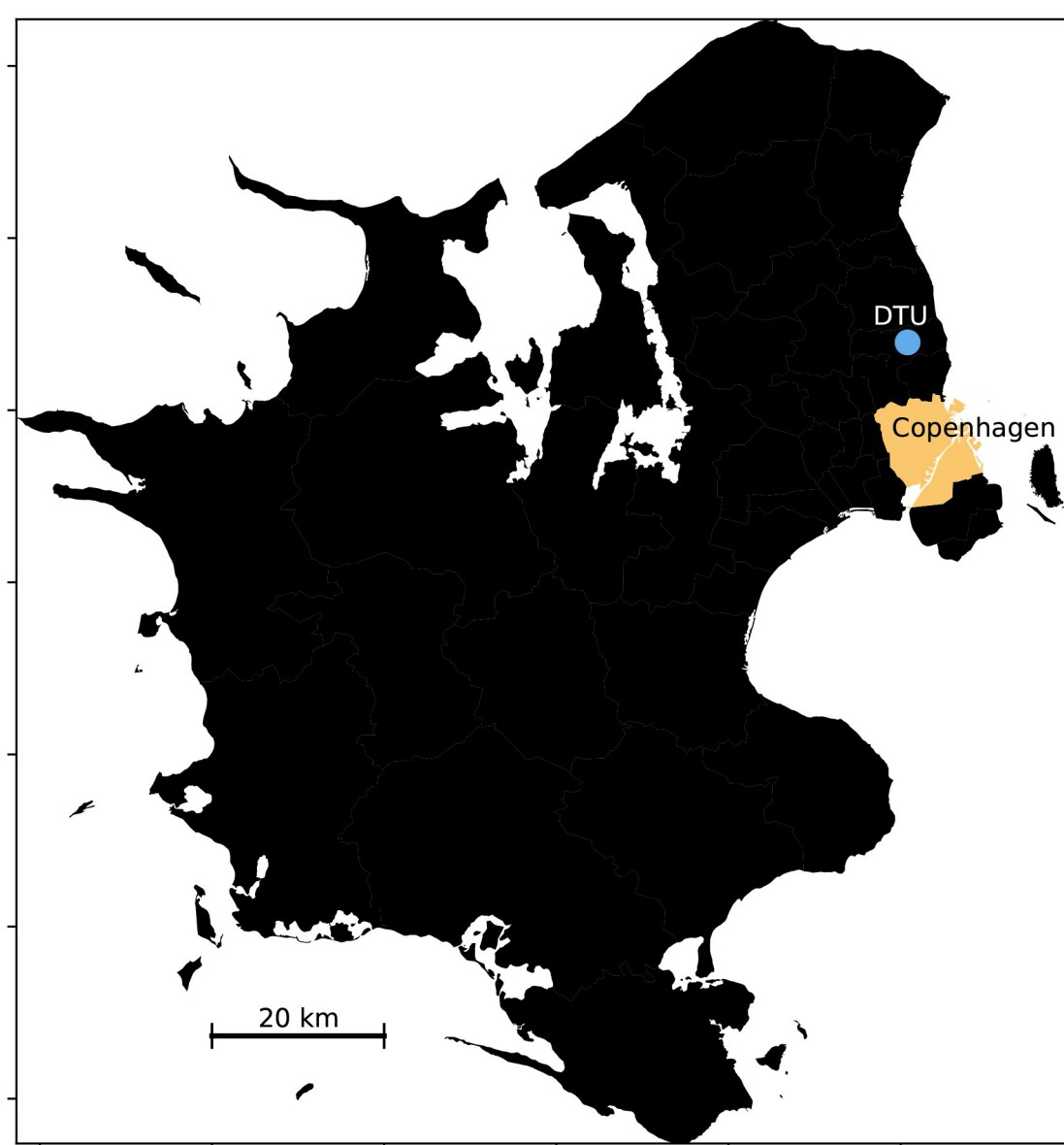

**Fig 2. Map over study area: Zealand, Denmark.** The study area spans Zealand, the largest island of Denmark which measures roughly 100 by 100 km. The light blue dot marks the Technical University of Denmark (DTU) and the area shaded in yellow demarcates Copenhagen, the Capital of Denmark.

Second, we extracted trip segments, i.e. time intervals where the users were moving. Third, we computed a number of features from the combined location data and raw Wi-Fi and Bluetooth data. In what follows we review details about each of these steps.

**Combined location data.** The geolocation of the study participants was approximated using a mixed approach in which we enriched low frequency location data from the Google Location API with additional Wi-Fi based location point estimations. We compute the combined location data by i) mapping out the location of routers (see below), ii) infer location from Wi-Fi data as median location of nearby routers, iii) merge the two data sources and resample as median location within every minute bin. After merging the location data from Wi-Fi, the temporal resolution of the Wi-Fi location data was one location point per minute

in places that were frequently visited. For seldom visited locations, the temporal resolution approximated one location sample every five minutes. Note that this lower frequency in temporal resolution is the result of missing Wi-Fi router location estimates in sparsely sampled regions.

The process of mapping out the Wi-Fi routers consisted of three main steps outlined in the following:

1. Estimate the locations of routers

    1. First, we remove potentially erroneous location estimations from users' traces using a speed-based correction method (eliminating jumps with speeds over $v_{max}$ = 180 km/h) similar to a common approach for correcting oscillation between multiple cell towers in CDR and sightings data research [84].

    2. Next, we extract 'stay periods' (periods with location updates at least every $t_{max}$ = 300 seconds and corresponding to location changes no larger than $d_{max}$ = 30 meters). We note that these thresholds were determined on the basis of the properties of the underlying data: $t_{max}$ = 300 s corresponds to the configured sampling period; $d_{max}$ = 30 m corresponds to the reported accuracy of location estimations from Google Location API; $r_{max}$ = 300 m corresponds to the maximum range of an off-the-shelf Wi-Fi router; $v_{max}$ = 180 km/h is a liberal limit of how fast one can move using the transportation modes we investigated. Previous research using Wi-Fi data collected the same way showed that mobility traces estimated using different thresholds do not differ significantly [85].

    3. Subsequently, we identify the approximate locations of scanned Wi-Fi routers by associating the Wi-Fi MAC-addresses to locations where they were scanned (both through a 'strict' approach where a location and a Wi-Fi scan happened simultaneously within the same second, and through a 'relaxed' approach where Wi-Fi scans happened within the previously extracted stay periods).

    4. We then compute the geometric median of all sightings for a router as an estimation of its location. During this step we also estimate noise (defined as the percentage of sightings at a distance greater than $r_{max}$ = 300 meters from the geo median).

    5. Finally, we remove Wi-Fi routers with noise above $n_{max}$ = 0.05 or with less than $s_{min}$ = 5 sightings.

2. Estimate the location of each scan by computing the geo-median location of all the scanned routers with known corresponding locations.

3. Combine Google Location API data with our Wi-Fi based estimations and remove jumps from the combined trace corresponding to speeds over 180 km/h or spatial precision exceeding 150m.

This method is based on previous work [66], but introduces two modifications: (1) we use all Google Location API data, not only GPS estimations, (2) we allow for sightings within stay periods rather than only allowing for sightings where the Google Location API estimation happened at the same second as a Wi-Fi scan. In this way, we are able to estimate the location of a larger fraction of routers without sacrificing the accuracy of the estimations.

**Segmentation.** When traveling from one location (home, work, etc.) to another, people often use more than one mode of transportation. Segmentation approaches enable improved estimates of transportation mode by breaking sequences of location points into smaller trip

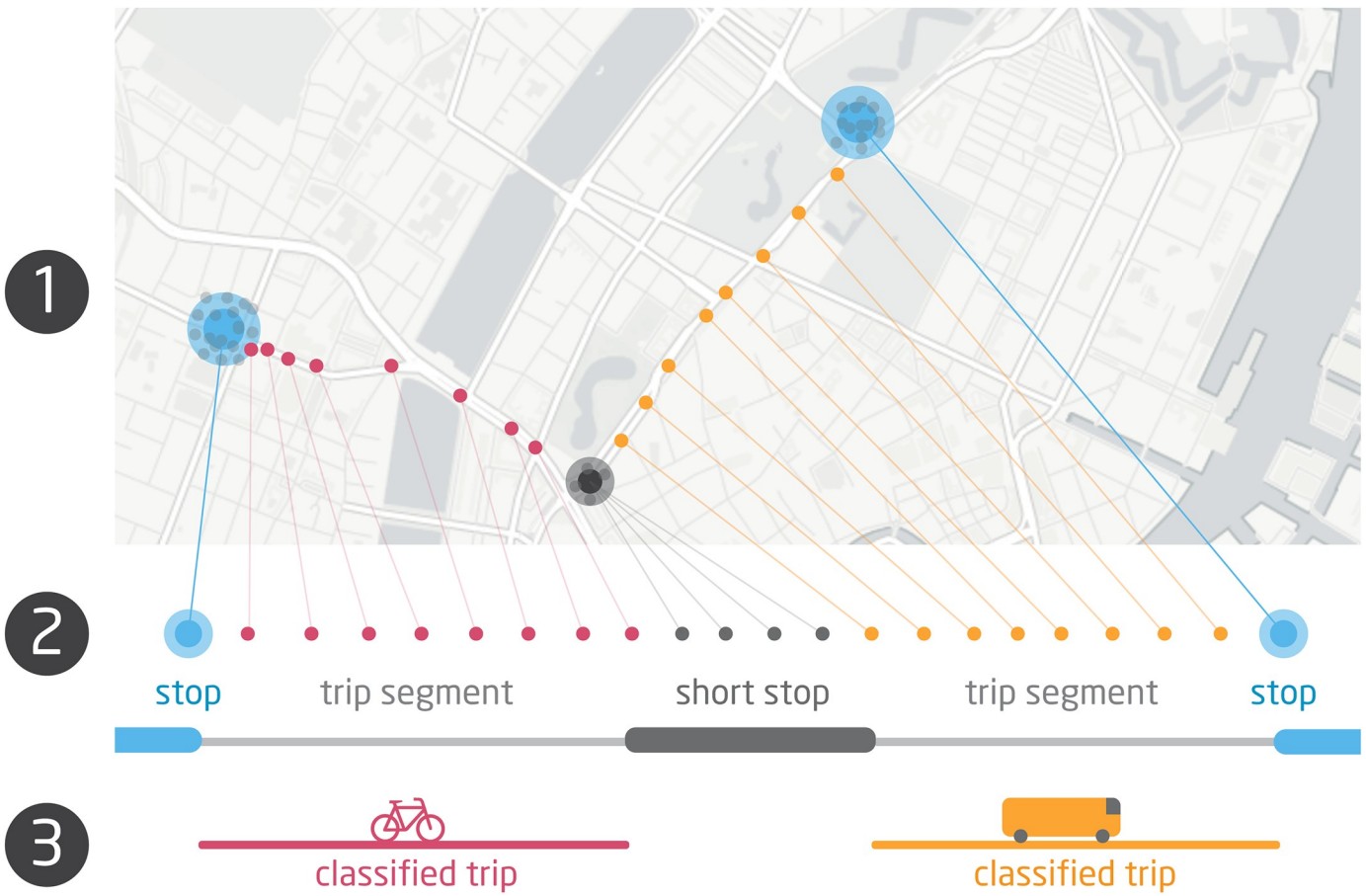

**Fig 3. Trip segmentation and feature extraction.** This figure highlights our segmentation and feature extraction strategy. In Step 1 the data collector app records GPS + Wi-Fi location traces. Step 2 segments trips using long (blue) and short stops (dark grey) and extracts features for training the model with labelled trips. Step 3 applies model predictions to classify trips with transportation mode.

segments, each corresponding to a unique mode of transportation [86]. Our segmentation strategy consists of two main steps, see Fig 3 for an overview of the process. We note that the estimated trip length varies across the segments, however, they are not used for inferring the mode.

First, we identify "long stops" (corresponding to home, work, etc.), where people dwell for an extended period of time, i.e. for a minimum of 15 minutes. These are used as the basis for initiating and completing sequential trips which may consist of multiple modes of transportation. In order to identify the long stops, we use the raw location data which is sampled at low frequency (every 5 minutes). Then we apply the stop detection algorithm outlined by [87]. We set the radius for categorizing a long stop to 200 m. By detecting when people stop at a particular location and dwell within the immediate vicinity for more than 15 minutes, and excluding these long stops from the transportation classification, we are able to constrain the classification to target people's transportation behavior from one long stop to the next. Our choice of 15 minutes as the minimum stay threshold in stops was chosen to be consistent with research on the stop location literature, see the review contained [88].

Second, we estimate "short stops". These are locations where people pause during their trips (e.g. waiting at a bus stop, grabbing a coffee to go, etc.) and may be transitioning between transit modes. In the case of short stops we use merged Wi-Fi geolocation data which has

higher temporal resolution (every 1 minute). Short stops are computed using the same procedure as above but only require that a person remain in the same place for more than 5 minutes (unlike the 15 minutes for long stops). This time length was chosen to accommodate the higher frequency of our combined location data and enable the detection of shorter stays in locations. The distinction between short and long stops prevented an issue with erroneously inferred stop locations for shorter stops as some routers changed location over the sampling period, likely because people moved homes.

After having estimated all stops we update the arrival and departure times of the long stops using the merged Wi-Fi geolocation data (with 1min. frequency).

**Extracted features.** Here we outline which features/variables that were computed and used as input in our transportation model. We describe the group of features associated with each of our data types below. Note that the pre-processed dataset with extracted features is available in S1 File and an overview of all the computed features is found in Table 3.

**Spatial features.** The first group of features are derived from using our combined measure of location data. The features are divided into "direct" and "indirect" where direct features consisted of users' speed and acceleration. We also computed indirect contextual measures for each user location using geospatial information about the public infrastructure of Zealand, which is the Danish island where the metropolitan area of Copenhagen is situated: i) we used OpenStreetMap data regarding the railway network to compute the distance to the nearest railway and metro line [89]; ii) we used transit feed data from "Rejseplanen", which is a publicly run travel planning site [90], to calculate the proximity to each bus route (within 5 km of the segment).

**Other features.** We further used data from the mobile phone Wi-Fi and Bluetooth sensors about recently scanned adjacent access points and Bluetooth-enabled devices to construct several novel features. These features included the total number of scanned access points within a minute, the number of scanned access points per scan, and the turnover in Wi-Fi APs/scanned Bluetooth devices (measured using Jaccard similarity). Since Wi-Fi AP names often contain useful information, we also collected the Wi-Fi SSIDs of nearby access points. Specifically, for each minute we counted the average number of scans containing one or more SSIDs that corresponded to publicly available Wi-Fi APs from either buses or trains. The collected SSIDs contained verified indicators of the associated mode of transportation, such as the previously

**Table 3. List of features computed.**

| Sensor | Feature |
|---|---|
| Bluetooth | unique_ids |
| | jaccard |
| Spatial | velocity |
| | acceleration |
| | train_dist_min |
| | metro_dist_min_cph |
| | bus_dist_min |
| Wifi | count_scans |
| | unique_ids |
| | empty_scans_share |
| | unique_routers_scan_mean |
| | jaccard |
| | bus_ssid_share_max |
| | train_ssid_share_max |

mentioned 'Bedrebustur'. Whenever one or more features were missing for a given minute, e.g. missing a scan or when the accuracy was below the previously stated threshold, we used linear interpolation to impute the values.

## 5 Results

### 5.1 Model performance

We now combine the elements discussed so far and apply machine learning to classify the collected data. The main focus of our analysis is to model the choice of transportation where we distinguish between {Car, Public, Self-powered}. To check whether the relative contribution of Wi-Fi features is robust across alternative sets of transportation modes, we also conduct our analysis on a larger set {Car, Train, Bus, Metro, Self-powered}. An overview of the performance for all the models we have used is found in S1 Table. In this section we focus only on results from Random Forest (RF) which performed the best of the considered models; see S2 Appendix for a detailed model comparison. Our random forest model yields a accuracy of 89% and $F_1 = 83\%$, see the overview of performance measures in Table 4.

During the model specification, we employed the RFE procedure to eliminate irrelevant features. In our model setup the number of features to keep with RFE was entered as a hyperparameter for optimizing in each iteration, see Methods for details. After fixing the number of features to use, the features with the lowest importance were excluded, see Section 3 for details. Overall, few features were eliminated from the base RF model, which included all input features; see S1 Fig. The plot shows that no features were eliminated in the majority of iterations ($> 80\%$) and two or more features were only removed in 2% of the iterations. S1 Fig also contains measures of feature importance which show that the spatial-based features, i.e. velocity and GIS, tend to rank highly. Some of the Wi-Fi and Bluetooth features, e.g. Wi-Fi context information for buses (called `bus_ssid_share_max`) have higher feature importance than the distance to nearest bus route (called `bus_dist_min`). For further inspection of the data see the conditional distribution of each feature by each of the three transportation modes in S2 and S3 Figs.

We emphasize that feature importance should only be seen as an informal ranking rather than a formal test of whether or not a certain group of features contribute to model accuracy at the segment level. The reason is that feature importance measures rely on minute level predictions where out-of-sample observations may be from the same trip segment, which means the predictions are likely to be too accurate and thus not trustworthy.

**Contribution of all Wi-Fi and Bluetooth features.** In order to formally test whether Wi-Fi and Bluetooth features contribute to the model, we computed the model performance at the

**Table 4. Performance measures for random forest model under various feature sets.**

| GIS context | Wi-Fi and Bluetooth features | Accuracy | $F_1$ score | Precision | Recall |
|---|---|---|---|---|---|
| Included | Included all | 0.893 | 0.825 | 0.863 | 0.802 |
| | Excluded all | 0.851 | 0.747 | 0.813 | 0.713 |
| | Excluded Wi-Fi context. | 0.862 | 0.766 | 0.813 | 0.739 |
| Excluded | Included all | 0.870 | 0.787 | 0.827 | 0.764 |
| | Excluded all | 0.763 | 0.563 | 0.640 | 0.545 |
| | Excluded Wi-Fi context. | 0.822 | 0.685 | 0.747 | 0.655 |

This table contains all the mean performance measures under various feature sets of including/excluding GIS features as well as including/excluding Wi-Fi and Bluetooth based measures. The mean is computed using the performance measures associated with each of the 1,000 resamples of the data, see Methods for details.

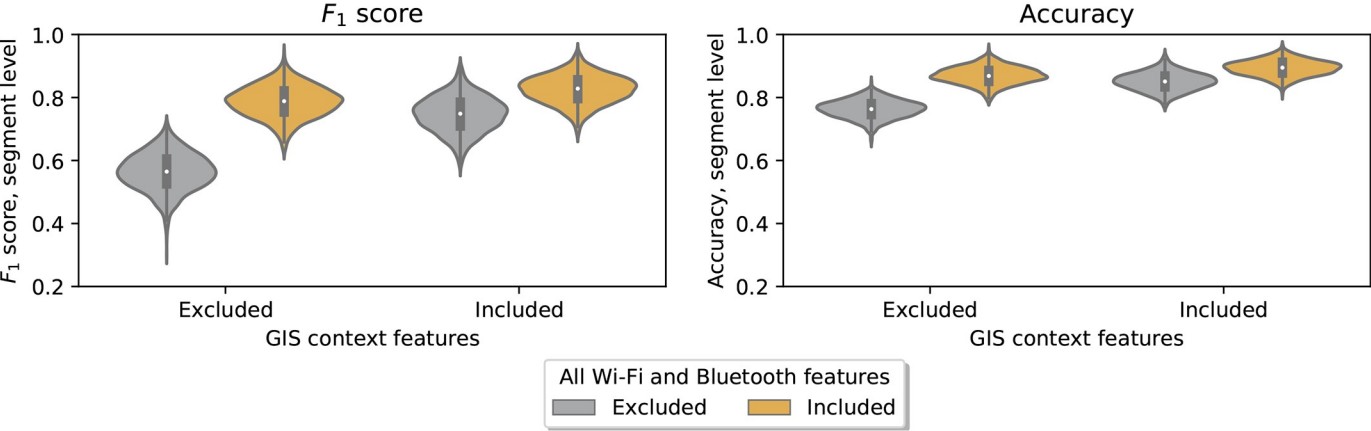

**Fig 4. Contribution of Wi-Fi and Bluetooth features to model performance, overall.** This figure shows the average model performance with and without all Wi-Fi and Bluetooth features. This difference is computed with GIS features excluded and included, respectively. The target is "Car vs. Public vs. Self-powered" and the performance measures are computed as mean across modes. The plots are created by resampling the data 1,000 times into test and training sets.

segment level when the set of Wi-Fi and Bluetooth features was separately included and then excluded. The inclusion of Wi-Fi and Bluetooth resulted in gains in $F_1$ score of respectively 0.223 ($p = 0.001$) without GIS features and 0.079 ($p = 0.006$) with GIS features, see Fig 4. These results demonstrate that the set of Wi-Fi and Bluetooth features contributes substantially to the model performance.

**Contribution of Wi-Fi context features.** We move on to analyzing whether the gains in model performance from Wi-Fi and Bluetooth features are driven by the Wi-Fi context features for bus and train, i.e. `bus_ssid_share_max` and `train_ssid_share_max`. Again, we evaluate how the inclusion and exclusion of these features affect the model performance at the segment level. Fig 5 shows the difference and performance from adding the Wi-Fi context features when the GIS features are included or not. The gains in $F_1$ scores are, respectively, 0.102 ($p = 0.001$) and 0.060 ($p = 0.009$) when GIS is excluded and included. Thus, adding the Wi-Fi context features set provides a significant contribution to the random forest model performance.

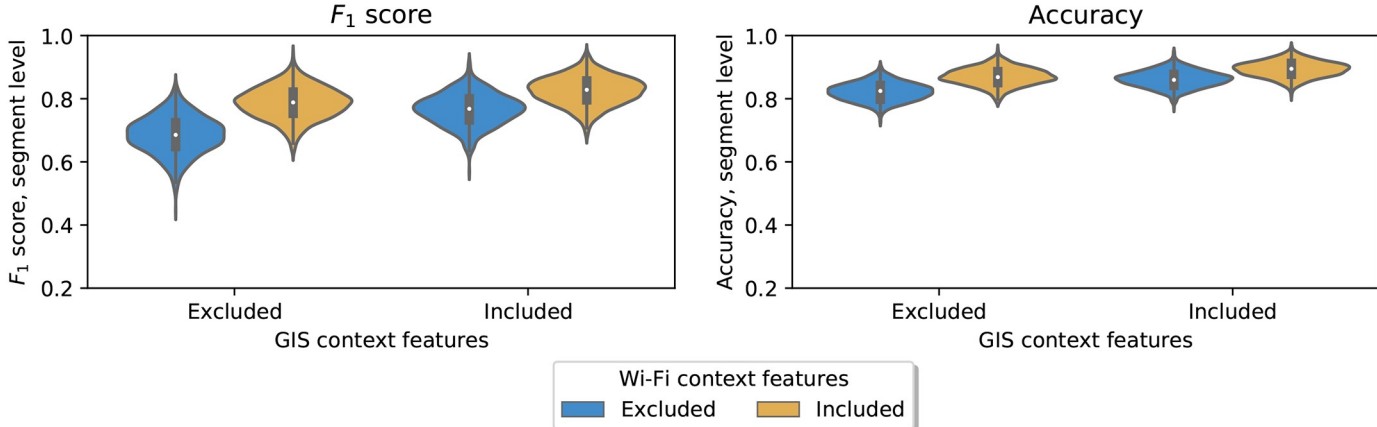

**Fig 5. Contribution of Wi-Fi context information to model performance, overall.** This figure shows the average model performance with and without the Wi-Fi contextual features for bus and train. This difference is computed with GIS features excluded and included, respectively The target is "Car vs. Public vs. Self-powered" and the performance measures are computed as mean across modes. The plots are created by resampling the data 1,000 times into test and training sets.

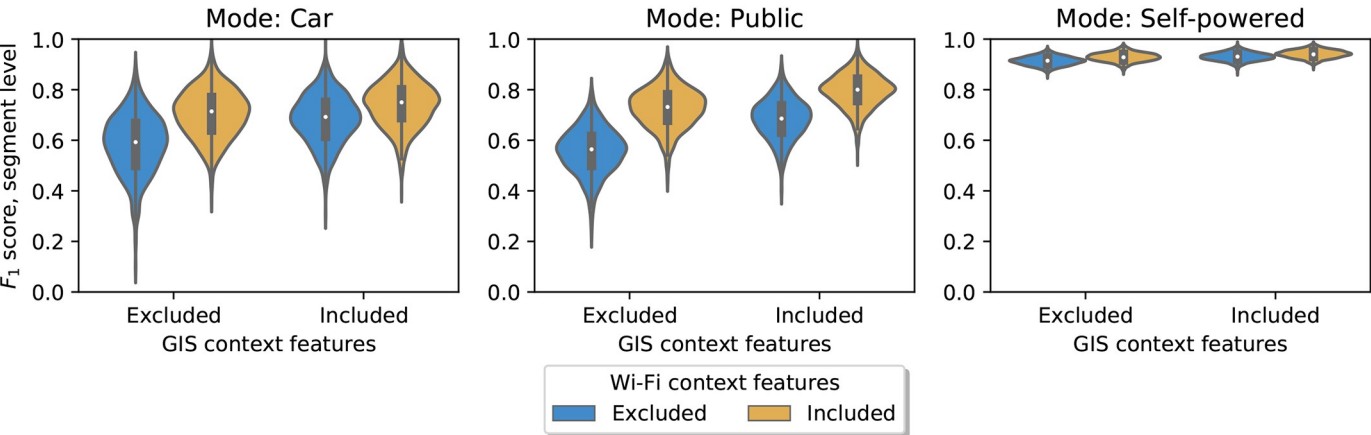

**Fig 6. Contribution of Wi-Fi context information to model performance, by mode.** This figure shows a breakdown of the model performance by mode. The model performances are computed when GIS as well as Wi-Fi and Bluetooth features are included and when they are excluded. The target is "Car vs. Public vs Self-powered". The plots are made from resampling the data 1,000 times into training data for model building and test data for evaluating the model.

Fig 6 shows a breakdown of differences with and without Wi-Fi context features. The figure shows that Wi-Fi features contribute large increases in $F_1$ for recognizing public transportation, marginal increases for cars and negligible changes for self-powered. The relative improvements in accuracy for all of these modes are greater when GIS features are not available, although a marginal increase in performance for detecting public transportation is still observable even when GIS features are present. An overview of the mean gains in performance by mode and associated p-values are found in S2 Table. The results show significant gains in model performance for public transport from adding the Wi-Fi context features both when GIS context features are included and when excluded. There are also modest gains for car and self-powered transportation.

**Robustness check: Increasing the number of modes.** We also find a positive contribution of Wi-Fi and Bluetooth features to model performance when changing the model target from 3 modes to 5 modes, see S2 Appendix. The gain in prediction performance relative to the model where all Wi-Fi and Bluetooth are excluded is 0.372 ($p = 0.001$) when excluding GIS features and 0.057 ($p = 0.000$) when GIS features are included. The gain relative to the model where only Wi-Fi context features are excluded is 0.144 ($p = 0.001$) without GIS features and 0.042 ($p = 0.09$) with GIS features. These results provide strong evidence that Wi-Fi and Bluetooth features meaningfully contribute to model performance when predicting 5 transportation modes.

**Robustness check: Lowering the sampling frequency.** To gauge the importance of the temporal resolution we ran an experiment where the models of three transportation modes were estimated at the 5-minute level instead of the 1-minute level, as we have done until now. The results are found in S2 Appendix. Compared with the main specification, the 5-minute model yielded a slightly higher $F_1$ score (0.85 up from 0.83) but slightly lower accuracy (0.88 down from 0.90) compared with the 1-minute model. Moreover, the results are qualitatively the same.

## 5.2 Model application to student mobile trace data

In order to visually inspect the predictive validity of how our model scales to a larger population, we apply our classification approach to mobile phone data collected from over 800 students who participated in the Copenhagen Networks Study (CNS) [71]. The CNS dataset

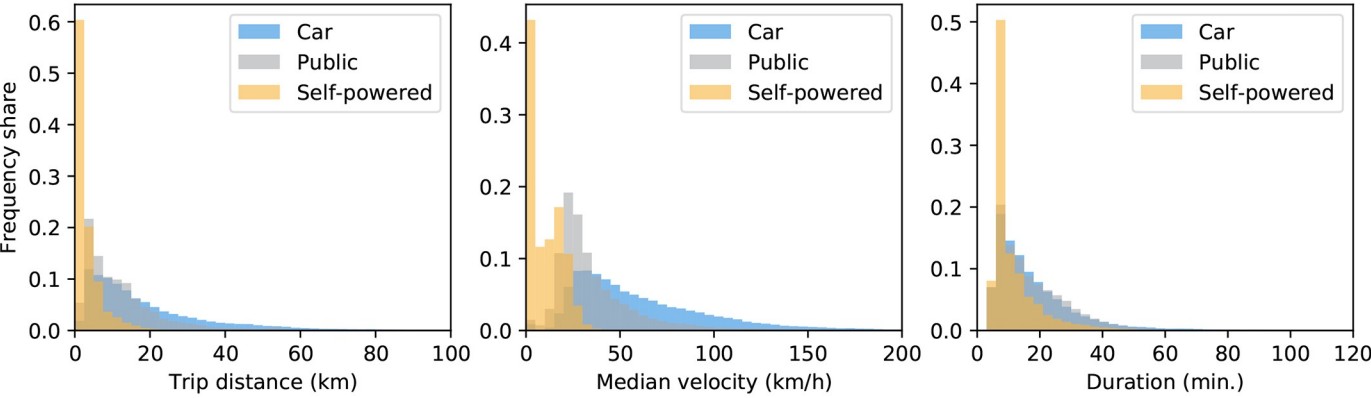

**Fig 7. Segment statistics for a large population.** (left panel) The distribution of trip distances for the three modes of transportation: Car, Public transport, Self-powered. (middle panel) Distribution of median trip velocity. (right panel) The distribution of trip durations. Taken together, the distributions reveal that applying our model to real-world data generates plausible population level statistics; see main text for details.

comprises both the geospatial traces and social interactions (calls, texts, Facebook, Bluetooth proximity) along with Wi-Fi scans for participating students enrolled at the Technical University of Denmark over a span of multiple years, and mirrors the location data quality and resolution collected in the current study. As mentioned earlier, the data collector app used by students did not collect and store accelerometer data, further motivating the present approach.

We filtered out users from CNS with less than two weeks of logged data. We then applied a model which combined 50 iterations of resampling of the random forest model (resulting in 5000 trees in total) to the data from the students. Using this model, we inferred the modes of transportation taken by 814 participants during a total of 669702 trips.

Analyzing trip level statistics for the inferred modes of transportation provides a "sanity-check" of the validity of the model. Fig 7 shows segment level statistics by inferred segment level transportation mode. The overall statistics reveal that the model generates plausible population-level statistics. Firstly, the distribution of velocity is as expected: lower for self-powered transportation and higher for car or public transport. Secondly, the travel time and distance also follow our expectations for Copenhagen: motorized transport is used for trips that last longer and cover longer distances, while the self-powered mode is primarily used for shorter journeys. Note that although the effect of velocity is mechanical, the travel time, for instance, is not a parameter in the classification.

We finish our analysis of student data by plotting inferred trips on maps. In Fig 8 there are three subplots—one for each transportation mode. It is evident that inferred car trips cover a greater variety of spatial trajectories throughout the study region than public and self-powered transport. By comparison, predicted public transport trips exhibit more spatially constrained and consistent trajectories, which match the public transport infrastructure around Copenhagen. Finally we note that self-powered trips demonstrate shorter spatial displacements as expected.

## 6 Conclusions

Even though large private organizations use smartphones to map nearby Wi-Fi access points and improve location positioning, see [91, 92], Wi-Fi and Bluetooth data have not been widely used by city officials and public researchers to improve the detection of how we get around in general and use public transportation modes in particular. In this paper we introduced a

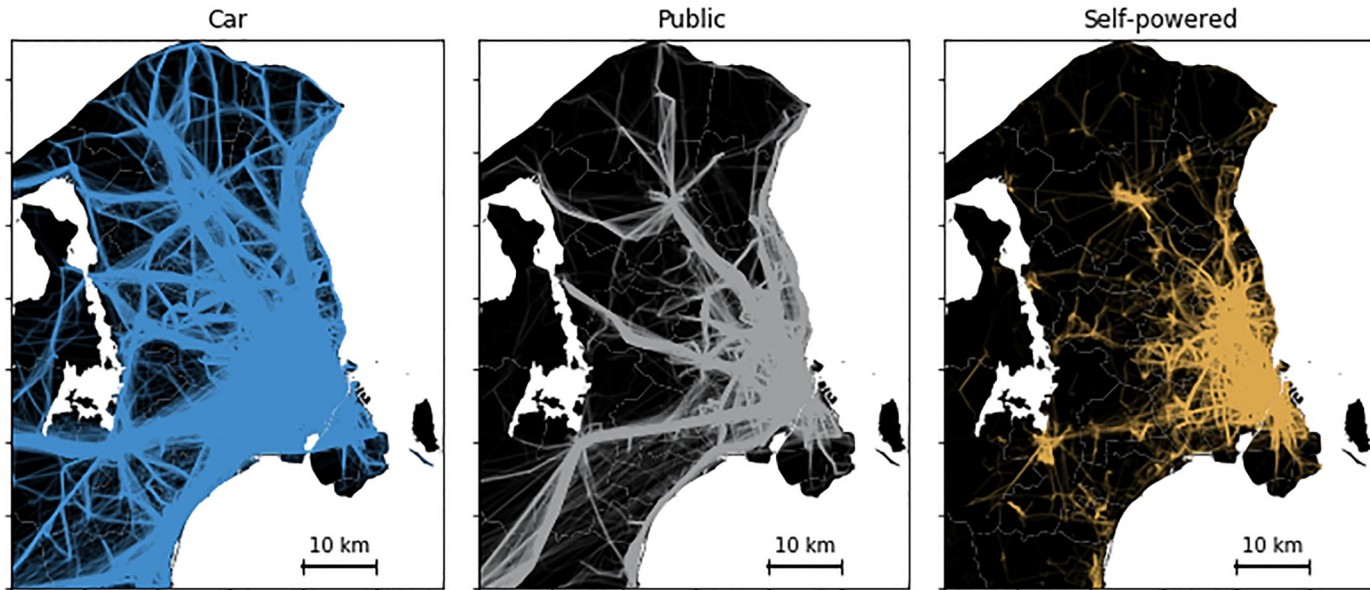

**Fig 8. Map of trips split by inferred mode for a large population.** The maps show trips by inferred mode around Copenhagen, Denmark. (left panel) Trajectories of students inferred to be using car; (middle panel) trajectories of student inferred to be using public transport; (right panel) trajectories of students inferred to be using self-powered transport.

method to leverage the contextual information embedded in now pervasive urban Wi-Fi and Bluetooth traces to help infer individual-level transportation mode choice.

We found that abundant Wi-Fi traces in connected urban environments and semantic clues embedded into public transportation Wi-Fi APs can directly disambiguate and help identify many commonly confused mode categories, including public transportation (bus, train, and metro) and cars. These novel features contribute considerably to model performance. Despite the comparatively sparse temporal resolution of our GPS and Wi-Fi data collection—and without the use of accelerometry—our random forest model yielded an overall accuracy of 89% and precision of 87% for classifying 3 transportation categories and separately, 5 different modes. This exceeds the performance of previous multi-mode classifiers incorporating Wi-Fi features [19, 24], is comparable to the performance of more recent low-power classification approaches [68, 70, 93], but performs worse than approaches that can afford higher temporal resolution GPS and/or accelerometer data [25, 94]. We investigated the predictive validity of our model using a large dataset which yielded predictions consistent with expectations of the different transportation modes.

Unlike accelerometer-based classification, Wi-Fi traces can provide both semantic clues about public transportation modes and the spatial position of the user. We showed that information from nearby Wi-Fi APs and Bluetooth-enabled devices collectively performs well as a viable substitute for computationally intensive GIS features, e.g. distance to bus routes. Checking whether Wi-Fi names contain the contextual information associated with the mode of transportation requires little computation and leverages mobile phones' existing Wi-Fi scanning activity. The gains in model performance from adding Wi-Fi context features for bus and train were not evenly spread across transportation modes and mainly helped to distinguish between car and public transportation. Public Wi-Fi continues to expand across both transportation modes and infrastructure, for instance in the U.K. [95]. Thus, Wi-Fi features represent an underutilized resource for making location based services for smartphones and smart cities that rely on accurate classifications of transportation mode.

Ours is the first study to demonstrate the viability of Wi-Fi features to contribute to the classification of public transportation and five common urban transportation modes. Our work is also the inaugural Wi-Fi feature-based classification study for the public transportation modes of bus, train, and metro. These modes are especially relevant for cities seeking to evaluate sustainable alternatives to car-based transportation and for transportation researchers seeking to distinguish the use of commonly confused modes from automated travel diary data. The ubiquity of smartphones that already collect proximate Wi-Fi access points at regular intervals and the relatively lower energy consumption of such sampling compared to GPS sensors [24, 67] support the viability of our method as an alternative, or supplement, to existing transportation mode detection algorithms.

## Limitations

Although our model has demonstrated promising initial performance, our study was limited in several important ways. Our training data collection was conducted by a small number of individuals from two different universities spanning the greater Copenhagen metropolitan area, with the University of Copenhagen located near the city center and the Technical University of Denmark located in a northern suburb. The sample consisted primarily of young students and researchers that represent a narrow temporal window over the human lifespan. Since this group is non-representative of the population's demographic diversity, self-powered mode velocities may have been biased towards the younger sample demographic. Another primary limitation of this study was that we did not collect concurrent accelerometer data during the training sessions and thus cannot conclude whether Wi-Fi features improve model performance beyond what is possible with accelerometry.

Additionally, this method relies on a dense network of encompassing Wi-Fi APs, ideally suited for urban areas. [53] and [96] have pointed out that the performance of network endpoint localization can diminish as the density of endpoints decrease towards recreational and/ or residential areas. Furthermore, another caveat of the current study is that contextual Wi-Fi SSIDs are not consistent between cities and thus need to be added and maintained for each desired urban context.

## Future research

Taxies, ferries, planes, ride-hailing services, and emergent forms of public transportation increasingly offer free Wi-Fi with transparent network names that provide public cues about the mode being used. Future research may seek to employ latent semantic information embedded into Wi-Fi APs in order to understand trip purposes, contexts and associated activities, building on the work of [97]. Although our approach has only been demonstrated in Copenhagen, a related research agenda might consider training an unsupervised classifier across several urban environments to identify mobile Wi-Fi APs that follow regular schedules and link these with known public transportation routes and schedules.

## Supporting information

**S1 Appendix. Method details.** This appendix contains information about our machine learning approach.
(PDF)

**S2 Appendix. Auxiliary results.** This appendix contains details of the analysis using other classification models and other transportation modes.
(PDF)

**S1 Fig. Feature importance and number of selected features.** These plots show the feature importance and number of features chosen for the optimal selected models for the 1,000 resampled models. We consider the random forest model, with all features; the target is "Car vs. Public vs. Self-powered". The black lines in the feature importance plots denote 95% confidence intervals based on resampled data. Feature importance values are computed for each random forest model at the temporal unit of observation (i.e. minute level) using Gini Impurity [79]. Note that feature importance is only for the random forest model with all features. The plots are created by resampling the data 1,000 times into training data for model building and test data for evaluating the model. Direct features are shown in blue, GIS features are shown in gray, and features based on Wi-Fi are displayed in yellow.
(PDF)

**S2 Fig. Geolocation features split by transportation mode.** This figure presents box plots for the geolocation based features. The plot contains both direct features, i.e. velocity and acceleration, and features based on distance to various geopgrahic information.
(PDF)

**S3 Fig. Wi-Fi and Bluetooth features split by transportation mode.** This figure presents box plots for the feature measured by Wi-Fi and Bluetooth features. The plot contains both features computed by investigating SSID for bus and train context as well as feature measuring the presence and change in Wi-Fi APs and Bluetooth devices.
(PDF)

**S1 Table. Overview of model performances.** This table contains all the mean performance metrics across all model specifications and feature sets investigates. The mean is computed using the resampled data 1,000 times into training data for model building and test data for evaluating the model.
(PDF)

**S2 Table. Overview of model performance gains.** This table contains all the gain in model performance broken down by mode when excluding either Wi-Fi context feature or all feature based on Wi-Fi or Bluetooth. Associated with the gains we have computed the $p$-value. The $F_1$ is a mean computed using the resampled data 1,000 times into training data for model building and test data for evaluating the model.
(PDF)

**S1 File. Files with dataset and code.** This compressed folder of files contains the dataset, code for training and evaluating models as well as model output.
(GZ)

## Author Contributions

**Conceptualization:** Andreas Bjerre-Nielsen, Kelton Minor, Sune Lehmann, David Dreyer Lassen.

**Data curation:** Andreas Bjerre-Nielsen, Kelton Minor, Piotr Sapieżyński.

**Formal analysis:** Andreas Bjerre-Nielsen.

**Funding acquisition:** Andreas Bjerre-Nielsen, Sune Lehmann, David Dreyer Lassen.

**Investigation:** Andreas Bjerre-Nielsen, David Dreyer Lassen.

**Methodology:** Andreas Bjerre-Nielsen, Kelton Minor, Sune Lehmann, David Dreyer Lassen.

**Project administration:** Andreas Bjerre-Nielsen, Kelton Minor, David Dreyer Lassen.

**Resources:** Kelton Minor.

**Software:** Piotr Sapieżyński.

**Supervision:** Sune Lehmann, David Dreyer Lassen.

**Visualization:** Andreas Bjerre-Nielsen.

**Writing – original draft:** Andreas Bjerre-Nielsen, Kelton Minor, Piotr Sapieżyński, Sune Lehmann, David Dreyer Lassen.

**Writing – review & editing:** Andreas Bjerre-Nielsen, Kelton Minor, Sune Lehmann, David Dreyer Lassen.

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
