## [Decision Letter · Decision Letter 0]

16 Jan 2020

PONE-D-19-33163

Inferring transportation mode from smartphone sensors: Evaluating the potential of Wi-Fi and Bluetooth

PLOS ONE

Dear Bjerre-Nielsen,

Thank you for submitting your manuscript to PLOS ONE. After careful consideration, we feel that it has merit but does not fully meet PLOS ONE’s publication criteria as it currently stands. Therefore, we invite you to submit a revised version of the manuscript that addresses the points raised during the review process.

We would appreciate receiving your revised manuscript by Mar 01 2020 11:59PM. To enhance the reproducibility of your results, we recommend that if applicable you deposit your laboratory protocols in protocols.io, where a protocol can be assigned its own identifier (DOI) such that it can be cited independently in the future. For instructions see: http://journals.plos.org/plosone/s/submission-guidelines#loc-laboratory-protocols

We look forward to receiving your revised manuscript.

Kind regards,

Lei Lin, Ph.D.

Academic Editor

PLOS ONE

Journal Requirements:

3. We note you have included a table to which you do not refer in the text of your manuscript. Please ensure that you refer to Table 1 in your text; if accepted, production will need this reference to link the reader to the Table.

Reviewers' comments:

Reviewer's Responses to Questions

**Comments to the Author**

1. Is the manuscript technically sound, and do the data support the conclusions?

Reviewer #1: Yes

Reviewer #2: Yes

2. Has the statistical analysis been performed appropriately and rigorously? 

Reviewer #1: Yes

Reviewer #2: Yes

3. Have the authors made all data underlying the findings in their manuscript fully available?

Reviewer #1: Yes

Reviewer #2: Yes

4. Is the manuscript presented in an intelligible fashion and written in standard English?

Reviewer #1: Yes

Reviewer #2: Yes

5. Review Comments to the Author

Reviewer #1: This paper uses information from Wi-Fi access points and Bluetooth devices, enhancing GPS and geographic data to improve transportation detection on smartphones through machine learning approaches. It is a timely study with promising future applications due to the prevalence of wi-fi, Bluetooth, and smartphones. The results are well explained. However, the methodology part is confused. There are some comments for authors to improve the paper.

1. In 2 Related Work section, for travel mode inferring, another type of research encompassing GIS application is using map service API, such as Google Route API, to provide API travel path/route for mode detection. As you mentioned Google Location API, more about those should be explored. You may refer to those research articles, and one example likes this:

a. Zhu, Lei, and Jeffrey D. Gonder. "A driving cycle detection approach using map service API." Transportation Research Part C: Emerging Technologies 92 (2018): 349-363.

2. In 3 Methods section, “We limited the number of classification models to three:” Why you choose those three methods? A little explanation may need here. Also, later, the random forest (RF) has been selected for analysis. A more detailed introduction of RF and its basis are necessary. I suggest adding some content about the RF basis.

3. In 3 Methods section, what are segment level and minute level? Please define them in the context. Also, why the segment level is selected?

4. In Figure 2, it is hard to read that DTU is an orange dot, not a blue dot. Also, for Copenhagen, it is in gray or orange?

5. On page 8, “The app collected location, Bluetooth, and Wi-Fi as detailed in [68]” Please explicitly explain the data collected. How does it look like roughly?

6. Page 9, “Subsequently, we identify router sightings, associating Wi-Fi routers to locations where they were scanned” What are router sightings? What is a Wi-Fi router? A picture or diagram of those or of the whole procedure of estimate the location routers may be helpful.

7. For segmentation on page 10, other literature on determining stationary time interval may need.

Reviewer #2: This paper studies an interesting problem of identifying the travel mode by using Wifi and bluetooth data. Comprehensive experiments have been conducted to show the effectiveness of using Wifi and bluetooth data for travel mode identification. However the motivations of choosing LR, SVM and RF are not clear. Why do you limit the classifiers to those three? The other concern is if the temporal resolution of the signal influence the performance. Some experiments should be performed to show the influence of the temporal resolution of the signal.

6. PLOS authors have the option to publish the peer review history of their article (what does this mean?). If published, this will include your full peer review and any attached files.

Reviewer #1: No

Reviewer #2: No

---

## [Author Response · Author response to Decision Letter 0]

11 Apr 2020

See attached document for a formatted "Response to Reviewers".

---

## [Decision Letter · Decision Letter 1]

18 May 2020

Inferring transportation mode from smartphone sensors: Evaluating the potential of Wi-Fi and Bluetooth

PONE-D-19-33163R1

Dear Dr. Bjerre-Nielsen,

We are pleased to inform you that your manuscript has been judged scientifically suitable for publication and will be formally accepted for publication once it complies with all outstanding technical requirements.

With kind regards,

Lei Lin, Ph.D.

Academic Editor

PLOS ONE

Additional Editor Comments (optional):

Reviewers' comments:

Reviewer's Responses to Questions

**Comments to the Author**

1. If the authors have adequately addressed your comments raised in a previous round of review and you feel that this manuscript is now acceptable for publication, you may indicate that here to bypass the “Comments to the Author” section, enter your conflict of interest statement in the “Confidential to Editor” section, and submit your "Accept" recommendation.

Reviewer #1: All comments have been addressed

Reviewer #2: All comments have been addressed

2. Is the manuscript technically sound, and do the data support the conclusions?

Reviewer #1: Partly

Reviewer #2: (No Response)

3. Has the statistical analysis been performed appropriately and rigorously? 

Reviewer #1: Yes

Reviewer #2: (No Response)

4. Have the authors made all data underlying the findings in their manuscript fully available?

Reviewer #1: Yes

Reviewer #2: (No Response)

5. Is the manuscript presented in an intelligible fashion and written in standard English?

Reviewer #1: Yes

Reviewer #2: (No Response)

6. Review Comments to the Author

Reviewer #1: All comments have been solved. Well done. No more comments. ..............................................

Reviewer #2: (No Response)

7. PLOS authors have the option to publish the peer review history of their article (what does this mean?). If published, this will include your full peer review and any attached files.

Reviewer #1: No

Reviewer #2: No

---

## [Editor Report · Acceptance letter]

3 Jun 2020

PONE-D-19-33163R1 

Inferring transportation mode from smartphone sensors: Evaluating the potential of Wi-Fi and Bluetooth 

Dear Dr. Bjerre-Nielsen:

I'm pleased to inform you that your manuscript has been deemed suitable for publication in PLOS ONE. Congratulations! Your manuscript is now with our production department. 

Kind regards, 

on behalf of

Dr. Lei Lin 

Academic Editor

PLOS ONE